# Prevalence and Genomic Characterization of Rotavirus A from Domestic Pigs in Zambia: Evidence for Possible Porcine–Human Interspecies Transmission

**DOI:** 10.3390/pathogens12101199

**Published:** 2023-09-27

**Authors:** Joseph Ndebe, Hayato Harima, Herman Moses Chambaro, Michihito Sasaki, Junya Yamagishi, Annie Kalonda, Misheck Shawa, Yongjin Qiu, Masahiro Kajihara, Ayato Takada, Hirofumi Sawa, Ngonda Saasa, Edgar Simulundu

**Affiliations:** 1Department of Disease Control, School of Veterinary Medicine, University of Zambia, Lusaka 10101, Zambia; atakada@czc.hokudai.ac.jp (A.T.); h-sawa@czc.hokudai.ac.jp (H.S.); nsaasa@gmail.com (N.S.); 2Laboratory of Veterinary Public Health, Faculty of Agriculture, Tokyo University of Agriculture and Technology, Saiwai-cho 3-5-8, Fuchu, Tokyo 183-8509, Japan; harima@go.tuat.ac.jp; 3Central Veterinary Research Institute (CVRI), Ministry of Fisheries and Livestock, Lusaka 10101, Zambia; hermcham@gmail.com; 4Division of Molecular Pathobiology, International Institute for Zoonosis Control, Hokkaido University, N20 W10, Sapporo 001-0020, Japan; m-sasaki@czc.hokudai.ac.jp; 5Division of Collaboration and Education, International Institute for Zoonosis Control, Hokkaido University, N20 W10, Sapporo 001-0020, Japan; junya@czc.hokudai.ac.jp; 6Department of Biomedical Sciences, School of Health Sciences, University of Zambia, Lusaka 10101, Zambia; anniekalonda@gmail.com; 7Hokudai Center for Zoonosis Control in Zambia, School of Veterinary Medicine, University of Zambia, Lusaka 10101, Zambia; misheckshawa@ymail.com (M.S.); kajihara@czc.hokudai.ac.jp (M.K.); 8Division of International Research Promotion, International Institute for Zoonosis Control, Hokkaido University, N20 W10, Sapporo 001-0020, Japan; 9National Institute of Infectious Diseases, Management Department of Biosafety, Laboratory Animal, and Pathogen Bank, Toyama 1-23-1, Tokyo 162-8640, Japan; 10Department of Virology-I, National Institute of Infectious Diseases, Tokyo 162-8640, Japan; 11Africa Centre of Excellence for Infectious Diseases of Humans and Animals, School of Veterinary Medicine, University of Zambia, Lusaka 10101, Zambia; 12Division of Global Epidemiology, International Institute for Zoonosis Control, Hokkaido University, N20 W10, Sapporo 001-0020, Japan; 13One Health Research Center, Hokkaido University, N18 W9, Sapporo 001-0020, Japan; 14Hokkaido University, Institute for Vaccine Research and Development (HU-IVReD), N21 W11, Sapporo 001-0020, Japan; 15Global Virus Network, 725 W Lombard Street, Baltimore, MD 21201, USA; 16Macha Research Trust, Choma 20100, Zambia

**Keywords:** rotavirus A, reassortment, interspecies transmission, genomic characterization, porcine, Zambia

## Abstract

Rotavirus is a major cause of diarrhea globally in animals and young children under 5 years old. Here, molecular detection and genetic characterization of porcine rotavirus in smallholder and commercial pig farms in the Lusaka Province of Zambia were conducted. Screening of 148 stool samples by RT-PCR targeting the VP6 gene revealed a prevalence of 22.9% (34/148). Further testing of VP6-positive samples with VP7-specific primers produced 12 positives, which were then Sanger-sequenced. BLASTn of the VP7 positives showed sequence similarity to porcine and human rotavirus strains with identities ranging from 87.5% to 97.1%. By next-generation sequencing, the full-length genetic constellation of the representative strains RVA/pig-wt/ZMB/LSK0137 and RVA/pig-wt/ZMB/LSK0147 were determined. Genotyping of these strains revealed a known Wa-like genetic backbone, and their genetic constellations were G4-P[6]-I5-R1-C1-M1-A8-N1-T1-E1-H1 and G9-P[13]-I5-R1-C1-M1-A8-N1-T1-E1-H1, respectively. Phylogenetic analysis revealed that these two viruses might have their ancestral origin from pigs, though some of their gene segments were related to human strains. The study shows evidence of reassortment and possible interspecies transmission between pigs and humans in Zambia. Therefore, the “One Health” surveillance approach for rotavirus A in animals and humans is recommended to inform the design of effective control measures.

## 1. Introduction

Rotavirus is a segmented, non-enveloped dsRNA virus of the family *Reoviridae*, and it is among the major causes of acute gastroenteritis (AGEs) in young children and animals globally [1,2]. Animals are considered to be among the natural reservoirs for rotavirus with rotavirus group A (RVA) being detected in asymptomatic animals such as bovine, porcine, and avian species [1]. Therefore, studying animal sources of zoonotic RVA is key to understanding the epidemiology of these viruses and developing preventive strategies in both animals and humans. In the livestock sector, rotavirus-associated infections result in major economic losses to farmers through high treatment costs, stunted growth, and varying mortality rates [3,4]. Ultimately, this leads to low productivity. The complexity of this problem leads to the empirical use of antibiotics since diagnostics are expensive, limited, or unavailable in most developing countries [5,6]. This may be contributing to the problem of antibiotic resistance in these countries. In Zambia, the livestock sector contributes 3.2% to the national gross domestic product (GDP) and 42% to the agriculture GDP [7]. Therefore, improved management of this sector could lead to economic development and improved livelihoods, particularly among farmers in rural areas.

In the human population, rotavirus caused a yearly estimate of 122,322–215,575 deaths in children under five years old before 2021 [8]. This is lower than the earlier estimate of more than 430,000 deaths per year before the introduction of rotavirus vaccines globally [9]. Africa alone accounts for about 50% of RVA-associated deaths globally despite the introduction of RVA vaccines in the immunization programs in most countries [9]. In Zambia, diarrhea-related deaths are ranked third among the major causes of death in children younger than five years [10].

The segmented nature of the RVA genome makes it highly prone to reassortment events [3,11]. In addition, co-infections and adaptability of RVA strains of different origins have been reported to be among the major causes of RVA reassortment [12]. Through these events, novel RVA strains emerge with the capacity to affect the effectiveness of current vaccines in pigs and humans [11,12,13]. Recent reports have shown that strains known to cause infections only in pigs are now being detected in humans with evidence of porcine–human reassortment [9,13,14]. Evidence also indicates that direct transmission of RVA from pigs to humans does occur [15,16]. For example, RVA strains of porcine origin have been detected in children with AGEs in Kenya, the Democratic Republic of Congo (DRC), Egypt, Thailand, Taiwan, China, and Argentina [15,17,18,19,20]. These data emphasize the need to understand the epidemiology of RVA of animal origin to assess their potential to spread in the human population.

In Zambia, like many developing countries, pigs are among the cheaper sources of protein. Therefore, the interaction of pigs and humans is unavoidable as they live in close proximity to humans, thus creating a public health risk. Hence, effective vaccines are the most effective way of preventing RVA-related infections in humans and animals [21]. Effective vaccines are strain-specific in animals, while in humans, they are not [22,23]. For example, currently approved vaccines for RVA in humans, like Rotarix, are derived from the G1 genotype, while RotaTeq is from genotype G1–G4 and a bovine genotype G6 virus. These two vaccines have been reported to reduce RVA infections in the human population even from infections associated with non-vaccine strains [22]. However, low efficacy of these vaccines has been reported in developing countries [23]. One of the proposed hypotheses for the observed low vaccine efficacy in these countries, besides nutrient deficiencies (zinc, vitamins A and D), breast milk antibodies, and environmental enteropathy, is the genetic diversity of RVA, which is associated mainly with reassortment and interspecies transmission between animals and humans [24]. Hence, understanding the epidemiology, evolution, ecology, and diversity of RVA in animals such as pigs is critical in generating valuable data that may be useful in explaining the low efficacy of these vaccines in developing countries, including informing vaccine design.

Rotavirus consists of 11 segments that encode six structural and six non-structural proteins, namely, VP1-VP4, VP6, VP7, and NSP1-NSP4, NSP5/NSP6 [25,26,27]. Currently, the rotavirus classification working group (RCWG) uses open reading frames to classify RVA based on the 11 segments as GX-P[x]-Ix-Rx-Cx-Mx-Ax-Nx-Tx-Ex-Hx [28]. Based on this, three major genotypes have been reported to cause infections in humans, namely, genotype 1, genotype 2, and genotype 3, referred to as Wa-like, DS-1-like, and AU-1-like, respectively [28]. The genetic backbone for genotype 1, genotype 2, and genotype 3 are I1-R1-C1-M1-A1-N1-T1-E1-H1, I2-R2-C2-M2-A2-N2-T2-E2-H2, and I3-R3-C3-M3-A3-N3-T3-E3-H3, respectively [29,30]. Interestingly, all these genotypes are believed to have an ancestral origin from animal sources (i.e., genogroup 1: porcine; genogroup 2: bovine; and genogroup 3: cats, dogs, and avian) [30]. To date, at least 42 G-types, 58 P-types, 32 I-types, 28 R-types, 24 C-types, 23 M-types, 39 A-types, 28 N-types, 28 T-types, 32 E-types, and 28 H-types have been detected (https://rega.kuleuven.be/cev/viralmetagenomics/virus-classification/newgenotypes, accessed on 25 June 2023). In addition, standardized naming criteria have been developed by the RCWG for RVA, which include the rotavirus group/species origin/country of identification/common name of the strain/year of detection/G/P types [28].

Host specificity for all the known RVA genotypes identified thus far has been described [31]. For instance, in humans, the predominant G-types are G1–G4, G9, and G12 in combination with the P-type P[4], P[6], and P[8], with G1P[8] and G9P[8] being the most common G–P combinations known to cause human infections [32]. In contrast, porcine G3–G5, G9, and G11, along with P[6], P[7], and P[8], with G5P[7] and G4P[8] are the most prevalent G–P combination [33]. In calves, the predominant genotype is G6 [34]. Recent studies from India, China, Argentina, Kenya, and the DRC have provided evidence of the new evolving genotype G4P[6] among the RVA strains causing severe diarrhea in young children globally [15,20,21,31,35,36]. Full genome analysis of the G4P [6] genotype from those studies revealed Wa-like and DS-1-like genetic backbones. These findings implied that the possible origin of G4P[6] would be porcine and bovine, respectively [18,20]. Also, whole genome analysis of a human G12 from Zambia, South Africa, Ethiopia, and Cameroon revealed a porcine–human reassortment [14]. However, in Zambia, G2 was the most common genotype in humans’ post-rotavirus vaccine introduction, and G1 and G9 were predominant during the pre-vaccination period [37]. Even though several genotypes have been detected in humans in Zambia, little is known about the origin of these genotypes. In animals, limited data are known about RVA in Zambia. To date, only three studies have been conducted, two in bats and one in rodents, and from these studies, novel genotypes were reported [38,39]. This implies that the diversity of RVA in the animal population in Zambia is highly probable. Therefore, surveillance of RVA in possible natural reservoirs of RVA such as pigs is of great importance to understand the possible source of existing and new emerging strains of RVA. Therefore, in this study, we screened stool samples from smallholder and commercial pig farms for RVA and conducted sequence analyses of the detected viruses to improve our understanding of the epidemiology and genetic diversity of RVA circulating in domestic pigs in Zambia.

## 2. Materials and Methods

### 2.1. Study Design and Study Site

We employed a cross-sectional study in selected farms in three districts of Lusaka city in the Lusaka Province of Zambia, namely, Chilanga, Kafue, and Chongwe (Figure 1). Our study sites included both commercial and smallholder farms. Farms with more than 100 pigs were considered commercial, and those with less than 100 pigs were categorized as smallholder farms. Sample collections were conducted during all three main seasons of Zambia, namely, wet rainy season from mid-November to April, hot and dry season from mid-August to mid-November, and cold and dry season from May to mid-August. We sampled piglets, weaners, and growers, regardless of whether they were symptomatic or asymptomatic, for diarrhea. Pigs less than 4 weeks, 4–8 weeks, and greater than 8 weeks old were considered piglets, weaners, and growers, respectively. Random sampling was conducted from January 2018 to December 2018.

### 2.2. Sample Collection, Preparation, and RNA Extraction

A total of 148 fecal samples were collected from pigs comprising 49 piglets, 89 weaners, and 10 growers from selected farms in peri-urban areas of Lusaka Province. From the 148 samples, 47 were obtained from asymptomatic pigs, while 101 were from pigs with diarrhea. The samples were transported on a portable 4 °C refrigerator to the virology laboratory at the University of Zambia, School of Veterinary Medicine. A 10% suspension of each sample was prepared in phosphate-buffered saline (PBS) and was centrifuged at 6000× *g* for 10 min. Then, 100 µL of the supernatant was used for RNA extraction using Trizol LS reagent according to the manufacturer’s instruction (Life Technologies Corporation, Carlsbad, CA, USA). Total RNA was eluted in 50 µL of RNase-free water and stored at −80 °C until use.

### 2.3. Genomic Screening for Group A Rotavirus

Nested polymerase chain reaction (PCR) was used for screening of RVA targeting the VP6 gene using primer pairs described previously [40]. A one-step reverse transcriptase PCR kit (Qiagen, Hilden, Germany) was used for the initial step under the following conditions: 30 min at 50 °C, 15 min at 95 °C, 45 cycles of 20 s at 95 °C, 30 s at 50 °C, 1 min at 72 °C, and a final extension step of 10 min at 72 °C. The first-round PCR products were used as templates for the second-round PCR using the *ExTaq* DNA polymerase according to the manufacturer’s instructions (Takara Biotechnology Inc., Shiga, Japan). The following thermal cycling conditions were used: 98 °C for 10 s, 35 cycles of 98 °C for 30 s, 54 °C for 30 s, 72 °C for 1 min, and a final extension at 72 °C for 5 min. Visualization of the PCR products was achieved using gel electrophoresis with ethidium bromide staining. The positives obtained from the initial screening for VP6 were later used for genotyping using the major outer surface protein VP7 using primer pairs as described previously [38]. VP7 PCR products were purified for Sanger sequencing using Wizard^®^ SV Gel and Clean-Up System (Promega, Madison, WI, USA) according to the manufacturer’s protocol. The purified DNA was then sequenced directly using a Big Dye terminator cycle sequencing ready reaction kit v3.1 on a 3500 Genetic Analyzer (Applied Biosystems, Foster City, CA, USA). Sequences generated were assembled and edited using GENETYX version 12 (GENETYX Corporation, Tokyo, Japan). All the nucleotide sequences obtained were deposited in Genbank under accession numbers OR294031 to OR294042.

### 2.4. Whole Genome Sequencing

Viral dsRNA in the positive fecal samples for the VP7 gene was isolated and enriched for whole genome sequencing. About 0.5–1.0 g of stool samples positive for the RVA genes were added to 10 mL of PBS and vortexed until well blended. The suspension was centrifuged at 10,000× *g* for 3 min, and the supernatant was filtered through a 0.45 µm membrane to remove unpelleted bacterial-sized substances. The filtrate was concentrated using Amicon Ultra-15 centrifuge filters (Merck Millipore, Darmstadt, Germany) according to the manufacturer’s instructions. The subsequent filtrate was used as material for RNA extraction with the QIAamp Viral RNA Mini Kit (Qiagen, Hilden, Germany) according to the manufacturer’s instructions. Then, viral dsRNA in extracted total RNAs were purified using the ISOVIRUS II kit (Nippon Gene, Tokyo, Japan) according to the manufacturer’s instructions. The purified dsRNA was used for cDNA synthesis. Double-stranded cDNA was synthesized using a Prime Script Double Strand cDNA Synthesis Kit (Takara Biotechnology Inc., Shiga, Japan) according to the manufacturer’s instructions. The cDNA libraries were prepared using Nextera XT DNA Library Preparation Kit (Illumina, San Diego, CA, USA) according to the manufacturer’s instruction and subsequently subjected to whole-genome sequencing on the Illumina HiSeq X Ten (Illumina way, San Diego, CA, USA). The raw sequence reads obtained were deposited in the National Center for Biotechnology Information (NCBI) database under the bioproject accession number PRJNA997783.

### 2.5. Sequence Data Analysis

Sequence data generated were produced at an average estimated depth of 37.7 from the Illumina HiSeq X ten and were analyzed using Geneious Prime^®^ version 2020.0.2. The workflow for the analysis of the imported FastQ files in Geneious prime was constructed as follows: trim leads (index), error correct and normalization, and lastly, assembly, which was conducted by mapping the normalized reads to RVA reference sequences downloaded from the GenBank. The consensus sequences generated were exported in Fasta files and genotyped using the Rotavirus Virus A Genotyping tool (https://www.rivm.nl/mpf/typingtool/rotavirusa), accessed on 21 June 2023. Phylogenetic trees of the 11 segments (VP1-VP4, VP6, VP7, NSP1-NSP5/6) of RVA were constructed using the maximum likelihood method in Molecular Evolutionary Genetics Analysis version 7 (MEGA 7) [41]. The best model for each tree was selected based on the Bayesian information score (BIC). The model with the lowest BIC score was chosen as the best substitution model for each tree. The trees generated included selected reference strains of RVA from pigs and humans with high nucleotide similarity to our study strains from Africa and the rest of the globe. The reference strains were identified using the Basic Local Alignment Search Tool (BLAST) (https://blast.ncbi.nlm.nih.gov/Blast.cgi, accessed on 25 April 2023). The nucleotide sequences that were shorter than the study strains were removed from the analysis. Nucleotide alignment was performed using Multiple Alignment Fast Fourier Transform (MAFFT) (https://mafft.cbrc.jp/alignment/software/), accessed on 20 January 2023. A bootstrap of 1000 replicates was used to determine the dependability of the branching order.

### 2.6. Statistical Analysis

Data collected regarding age, seasons, and health status (diarrhea) from pig farms were analyzed using the R package dplyr v1.0.7 (RStudio, Boston, MA, USA) and visualized using ggplot2 v3.3.5 (Houston, TX, USA). Furthermore, data analysis was conducted in R, with statistical significance set at *p* ≤ 0.05. Specifically, univariate logistic regression was performed to select which variables to use in the multiple regression analysis, where variables with a *p*-value less than 0.2 qualified for analysis. Finally, the coefficients of the variables in the univariate and multivariate regression models were exponentiated to obtain odds ratios.

## 3. Results

### 3.1. Detection of Group A Rotaviruses

We screened a total of 148 samples for the presence of RVA using nested PCR, targeting segment 6 encoding the VP6 gene. From these, 34 were positive on PCR, comprising 6 from asymptomatic and 28 from symptomatic pigs, representing a prevalence of 22.9% (34/148). Across age groups, the 34 positive samples were distributed as follows: piglets (*n* = 18), weaners (*n* = 15), and growers (*n* = 1). In addition, there were more positive samples from commercial farms (*n* = 33) than from smallholder farms (*n* = 1). Using binary logistic regression, we modeled the dependence of RVA infections on different variables. First, we performed a univariate screening using *p* = 0.2 as our cut-off point (Table 1). The results revealed that all variables (i.e., season, diarrhea, and age) had a *p*-value below the set threshold (i.e., *p* < 0.2) and thus qualified for inclusion in the multivariate regression analysis (Table 2). The result of the final model showed that only season was significantly associated with RVA infections (odds ratio (OR) = 3.13, 95% confidence interval (CI) = 1.37–7.47, *p*-value = 0.008). However, diarrhea was not significantly associated with RVA infections (*p* > 0.05) (Table 2). Similarly, age (grower, piglets, and weaners) had no significant association with RVA infections (*p* > 0.05) (Table 2).

### 3.2. Genotyping of Group A Rotavirus Using VP7 Gene

Thirty-four stool samples positive for RVA on the VP6 gene were further screened for the VP7 gene by PCR to determine the G-genotypes of each positive sample. Out of these, 12 were positive for the VP7 gene and were all successfully Sanger-sequenced. Thereafter, BLASTn was used to assess the nucleotide similarity with those in the National Center for Biotechnology Information (NCBI) database. Out of the 12 positives, six had high nucleotide sequence similarity to porcine RVA strains and the other six to human RVA strains, with the percentage identity ranging from 87.5% to 97.1%. Notably, from the 12 positives for the VP7 gene, only one sample was recovered from asymptomatic pigs, and 11 were from symptomatic pigs. BLASTn also revealed three distinct G-genotypes that had high nucleotide sequence similarity to our study sequences, namely G5 (*n* = 5), G9 (*n* = 6), and G4 (*n* = 1) (Table 3). It was noted that some strains that showed high sequence similarity to our study samples in GenBank were isolated from pigs, while others were detected in humans (Table 3).

### 3.3. Whole Genome Sequence Analysis and Genetic Constellation Determination

From the 12 samples that were positive for the VP7 gene, six were selected for whole genome analysis based on PCR product intensity and the RNA concentration of the sample. Out of the six samples that were subjected to full genome sequencing, we obtained the two full-length genomic sequences of samples, LSK0137 and LSK0147, with good reads of 5.1 × 10^7^ and 6.6 × 10^7^, respectively. The rest of the samples yielded few reads and had poor coverage. Therefore, we genotyped these two strains (LSK0137 and LSK0147) using the Rotavirus Virus A Genotyping tool (https://www.rivm.nl/mpf/typingtool/rotavirusa, accessed on 21 June 2023) and their genetic constellation was found to be G4-P[6]-I5-R1-C1-M1-A8-N1-T1-E1-H1 and G9-P[x]-I5-R1-C1-M1-Ax-N1-T1-E1-H1, respectively. However, for sample LSK0147, only nine segments were genotyped; the other two segments were short and did not meet the minimum% score for assigning genotypes (Table 4) [30].

Interestingly, out of the 11 segments of sample LSK0137, BLASTn assessment showed that nine had nucleotide sequence similarity to human RVA, and two were similar to porcine RVA, with the percentage identity ranging from 95.2% to 98.8% (Table 4). Even though we did not genotype all the segments for LSK0147, all 11 segments were assessed on BLASTn. The results showed that out of 11 segments, six showed the highest nucleotide sequence similarity to human strains, and five were similar to porcine RVA, with the percentage identity ranging from 93.2% to 99.6%.

### 3.4. Phylogenetic Analysis of RVA Structural Proteins Genes

Phylogenetic analysis of segment 9 encoding the VP7 gene revealed that LSK0137 and LSK0147 belonged to G4 and G9 genotypes, respectively (Figure 2). Segment 9 of LSK0137 was closely related to the RVA strain from China and distantly related to an RVA strain detected in Sri Lanka, while LSK0147 grouped with viruses obtained from pigs in Taiwan (Figure 2). Segment 6 of LSK0137 clustered with porcine RVA, which was detected in Vietnam, while LSK0147 clustered separately. However, both belonged to genotype I5 viruses (Figure 3). Phylogenetic analysis of segment 4 of LSK0137 revealed that it belonged to genotype P[6], and it was closely related to a human RVA, which was detected in China in 2018, while LSK0147 grouped within a clade that was composed predominantly of strains of porcine origin (Figure 4).

The sequences of segment 1 encoding the VP1 gene of LSK0137 and LSK0147 clustered with strains detected in pigs and humans from Africa and Asia. The VP1 sequence of LSK0137 was closely related to a porcine RVA strain from Mozambique, while that of LSK0147 was closely related to a human RVA strain detected in Thailand (Appendix A). Phylogenetic analysis of the VP2 gene of LSK0137 showed that it belonged to a cluster of RVA strains detected from pigs and humans in China, while the sequence of LSK0147 clustered distinctly (Appendix A). The virus sequence of segment 3 encoding the VP3 gene of LSK0137 grouped distinctly, while that of LSK0147 clustered with RVA strains of porcine origin from Mozambique and Vietnam (Appendix A).

### 3.5. Phylogenetic Analysis of RVA Nonstructural Proteins Genes

Phylogenetic analysis of segment 5 encoding the NSP1 gene revealed that LSK0137 belonged to genotype A8 and clustered in a clade with human RVA strains detected in Thailand and one porcine RVA from Vietnam, while LSK0147 was closely related to a human strain from China (Appendix A). Analysis of the NSP2 gene sequence of LSK0137 and LSK0147 showed that they belonged to genotype N1 but were grouped in separate clusters. The NSP2 gene sequence of LSK0137 was closely related to human RVA strains from China, while that of LSK0147 was closely related to porcine RVA from DRC (Appendix A). The analysis of the NSP3 gene virus sequences showed that both LSK0137 and LSK0147 belonged to genotype T1, and they were closely related to porcine RVA from China and Sri Lanka, respectively (Figure 5). Analysis of the enterotoxin NSP4 gene of LSK0137 and LSK0147 revealed that they belonged to genotype E1 (Figure 6). The sequence of LSK0137 belonged to a clade that included RVA strains from pigs found in Ghana, Mozambique, and China, while LSK0147 was closely related to RVA detected in pigs from Vietnam and was within a clade that had both pigs and human-origin strains (Figure 6). Phylogenetic analysis of the NSP5 gene showed that LSK0137 belonged to the H1genotype (Appendix A), and it clustered with RVA strains of pig origin, while LSK0147 was closely related to RVA obtained from a pig in Belgium. On the other hand, LSK0147 was closely related to a human strain from Sri Lanka (Appendix A).

## 4. Discussion

In the present study, we report the detection and genetic characterization of RVA in domestic pigs in the Lusaka Province of Zambia. We screened a total of 148 samples, out of which 34 samples were positive for RVA, representing a prevalence of 22.9% (34/148), which is within the reported global prevalence of RVA in pigs of 20% to 60% [33]. Further screening of the positives with VP7 primers produced only 12 positives representing the prevalence of 8.1% (12/148). This reduction may have been due to differences in the sensitivity between VP6 and VP7 primer sets that were used in this study. Alternatively, primers targeting the VP4 gene could have been used as they offer the advantage of determining the P genotype of detected RVA. In Africa, the overall prevalence of porcine RVA infection is unknown due to limited studies [33]. However, the prevalence of RVA infection in pigs in some African countries has been reported, for example, Kenya and Uganda (26.2%), Tanzania (35.3%), Mozambique (11.8%), Ghana (10.4%), and South Africa (27.3%) [3,4,12,13,42]. Based on these prevalence trends, of which more than 50% of what has been reported is above 20%, suggest that the overall prevalence of porcine RVA infections in Africa might be within the reported global prevalence [33].

Our study also showed that the rainy season is one of the major risk factors associated with RVA infections. This may have been associated with poor drainage systems in most farms and low temperatures that characterized the rainy season [43]. However, more studies are needed to explore the association of RVA infections and seasonality in pigs in Zambia and how this may affect RVA infection in humans. This finding suggests that stringent measures should be employed during the rainy seasons to curtail the spread of RVA. Though many positives were obtained in piglets and weaners when compared to growers, the uneven number of samples collected across the age groups may have influenced the results. Despite the differences in the sample numbers collected across age groups, our results of a higher detection rate in piglets and weaners than growers agree with what has been reported previously [12]. However, further studies are needed to explore the association between RVA infections and age in pigs in Zambia. Also, it would be interesting to compare the prevalence rates between commercial farms and smallholders in future studies. Additionally, the insignificant association of RVA infections to diarrhea in pigs adds to the existing knowledge that pigs may be among the natural reservoirs of RVA and may explain why RVA is detected in asymptomatic pigs [44]. Hence, more surveillance is vital in pigs to understand the epidemiology and diversity of porcine RVA in Zambia.

Further, BLASTn of the VP7 gene produced a low overall percentage identity with the reference strains from Genbank, suggesting that these may be unique strains [3]. Interestingly, some of the identified genotypes were similar to human strains and others to porcine strains. Therefore, the results may suggest possible interspecies transmission of RVA between pigs and humans [1]. Moreover, the circulation of these genotypes in pigs may be the source of new strains in the human population and may affect the effectiveness of both pig and human vaccines [45]. Comprehensive genetic characterization of RVA in animals such as pigs is vital in detecting variants that may be involved in interspecies transmission of RVA. For example, strains known to cause porcine infections have been detected in children with AGE in Egypt, Cameroon, and Nigeria [15,16,46]. This may suggest that reassortment events and interspecies transmission of RVA may be the source of these strains in the human population.

We obtained the whole genetic constellation of the LSK0137 strain and a near full-length genetic constellation of the LSK0147 strain, both with a Wa-like genetic backbone, as G4-P[6]-I5-R1-C1-M1-A8-T1-E1-H1 and G9-P[x]-I5-R1-C1-M1-Ax-T1-E1-H1, respectively. Though we did not manage to obtain the full genome of LSK0147, the two missing segments, VP4 and NSP1, showed highest nucleotide similarity of 93.2% and 98.2% to P[13] and A8 genotypes, respectively, when analyzed by BLASTn. Thus, the genotypes of VP4 and NSP1 of LSK0147 strains are most likely P[13] and A8 genotypes, respectively. Hence, the most probable genetic constellation of the LSK0147 strain would be G9-P[13]-I5-R1-C1-M1-A8-T1-E1-H1. Therefore, the Wa-like genetic backbone of our study strain suggests that their origin may be from pigs. This is because the Wa-like genetic backbone is associated with porcine origin [30]. We propose the name for porcine RVA in Zambia for LSK0137 and LSK0147 according to RCWG as RVA/pig-wt/ZMB/LSK0137/2018/G4P6 and RVA/pig-wt/ZMB/LSK0147/2018/G9P[13], respectively [28]. Analysis of both strains revealed some segments with genotypes similar to human and pig strains on BLASTn within the same genetic constellation. This is suggestive of the possible occurrence of reassortment events in the study strains.

Notably, reassortment events are known to be among the primary contributors to new evolving strains of RVA in animals and humans [13]. Recent studies from animals in Zambia revealed novel RVA genotypes [39]. Therefore, the detection of reassortant strains in pigs reared in close proximity to humans emphasizes the importance of surveillance in animals like pigs to avoid spillover of such strains in the human population. For example, rare genotypes like G4 with features of reassortment within its genetic constellation have been detected in Kenya and the DRC, and their genetic characteristics were traced to porcine [3,6]. These reports indicate that reassortment might be playing a role in the evolvement of new strains of porcine RVA with the potential to infect humans. While our results suggest RVA interspecies transmission, viral isolation-based RNA screening may be recommended to fully assess the zoonotic potential of these reassortant RVAs circulating in the pig population [39].

Phylogenetic analysis of the VP7 gene of the LSK0137 strain revealed that it belonged to the G4 genotype, and its origin may be from pigs as it formed a cluster with porcine and human–porcine reassortant RVA from China and Sri Lanka, respectively. The relatedness of the VP7 gene with reassortant strains may suggest that genotype G4 of LSK0137 may be zoonotic [47]. However, the VP7 gene of the LSK0147 strain grouped with porcine-origin strains from Taiwan belonging to the G9 genotype, suggesting that its origin would be from pigs. Analysis of the VP6 gene of the LSK0137 and LSK0147 strains showed that they belonged to genotype I5, which is mostly associated with RVA of porcine origin. This implies that their ancestral origin would possibly be pigs. The analysis of segment 4 (VP4 gene) of the LSK0137 strain, which belonged to the P6 genotype, showed that it was closely related to a human strain from China. This finding suggests possible interspecies transmission of RVA between pigs and humans. On the other hand, analysis of segments 1, 2, and 3 (VP1, VP2, and VP3) of the LSK0137 and LSK0147 strains revealed that these segments might be of porcine lineage, as they mainly clustered with strains from pigs. However, the origin of the VP2 gene of the LSK0147 strain and the VP3 gene of the LSK0137 strain could not be ascertained, as they clustered separately. This may imply that the genotypes of these segments may be new [12]. In addition, the relatedness of some of these segments with those from humans may imply that reassortment between pig and human RVAs does occur. Analysis of all the non-structural proteins (NSP1, NSP2, NSP3, NSP4, and NSP5/6) of the LSK0137 and LSK0147 strains showed a similar pattern with the structural proteins, as they mainly clustered with strains from pigs and some from humans. This suggests that the segments encoding the above-mentioned non-structural proteins may originate in porcine. The overall analysis of all 11 segments of RVA of the LSK0137 strain and some segments of the LSK0147 strain revealed that their likely origin would be porcine. However, the relatedness of some segments to human strains might suggest that interspecies transmission and reassortment may be occurring between pig and human RVAs in Zambia [15]. Though our study seems to suggest that LSK0137 and LSK0147 emerged in pigs, the limited number of full genomes analyzed, lack of samples from other animals such as cattle and humans, and the limited geographical coverage of sampling sites make it difficult to fully comprehend the origin and molecular epidemiology of RVA in mammalian hosts (including humans) in Zambia. However the high sequence similarity and close phylogenetic relationships of some gene segments of viruses characterized in this study to those viruses detected in humans suggest that pigs might be among the sources of the unusual RVA emerging in humans like G4P[6] in Africa [3,12,16]. Thus, there is a need for continued surveillance of both pigs and humans to ascertain the possibility of interspecies transmission of RVA in Zambia.

## 5. Conclusions

This study has shown for the first time evidence of the circulation of porcine RVA in Zambia. Nucleotide sequence and phylogenetic analysis suggest possible reassortment and interspecies transmission of RVA involving pigs and humans. Further, the insignificant statistical association of RVA infections with diarrhea supports the idea that pigs may be among the natural reservoirs of RVA in Zambia. Therefore, genomic surveillance of RVA in humans, pigs, and other animals is recommended for a better understanding of the epidemiology of RVA in Zambia.

## Figures and Tables

**Figure 1 pathogens-12-01199-f001:**
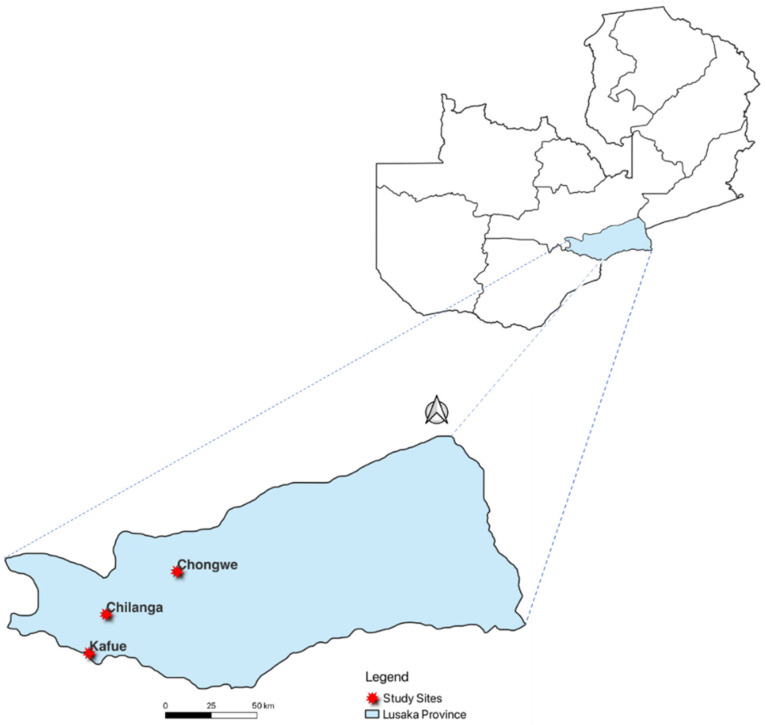
Map of Zambia showing Lusaka Province and districts of sample collection.

**Figure 2 pathogens-12-01199-f002:**
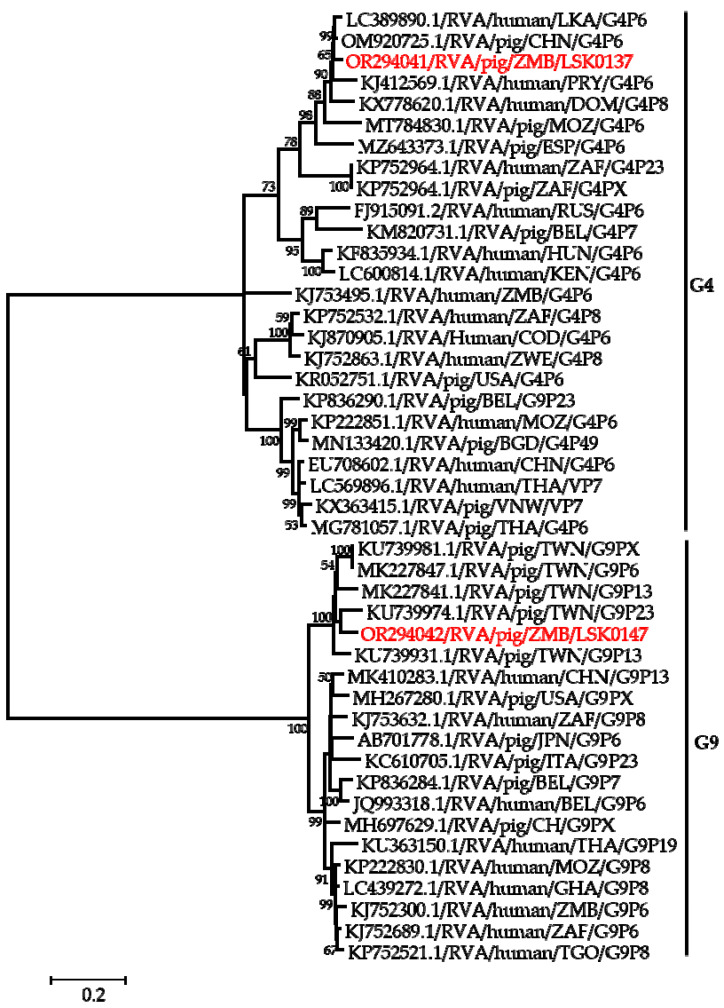
Phylogenetic tree of VP7 genes. The tree was constructed using the general time reversible evolutionary model. The analysis was based on 784 nucleotides. Bootstrap values on branch nodes ≥ 50% are shown. The GenBank accession number/rotavirus group/species origin/country of origin/G/P-types represent the reference sequences included in the tree. The viruses characterized in this study with their accession numbers are shown in red text. The vertical lines represent genotypes.

**Figure 3 pathogens-12-01199-f003:**
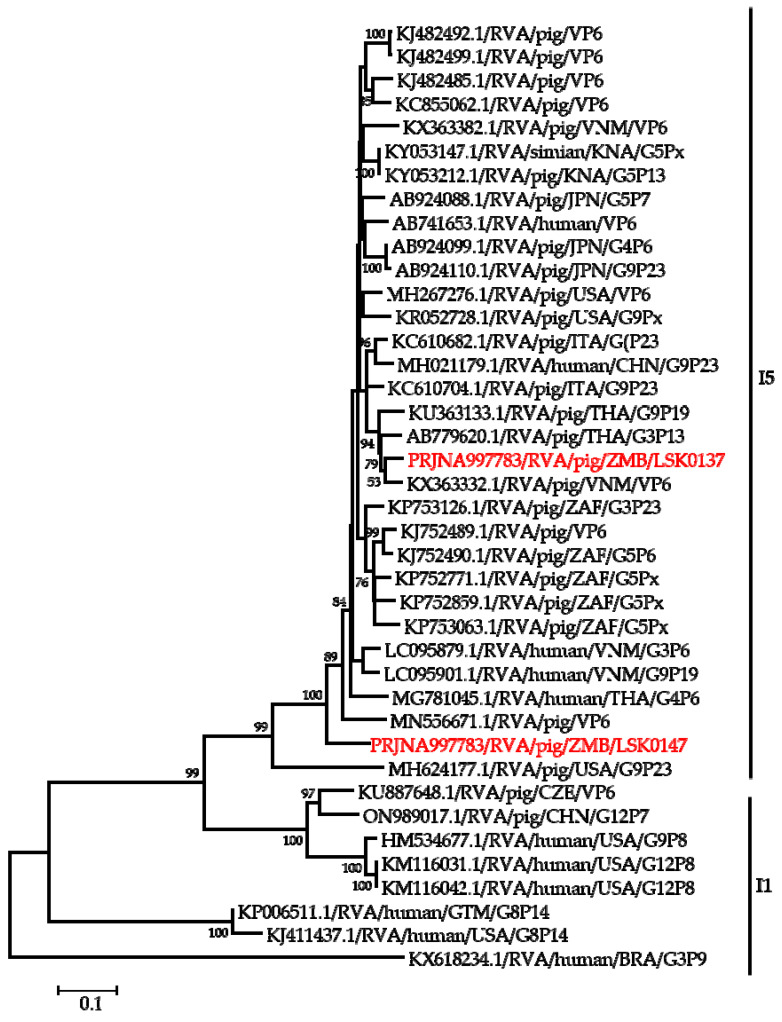
Phylogenetic tree of VP6 genes that belonged to I5 genotype. The tree was constructed using the maximum likelihood method based on the general time reversible with gamma distribution with some invariable sites (GTR + G + I) evolutionary model. The analysis was based on 1222 nucleotides. Bootstrap values on branch nodes ≥ 50% are shown. The GenBank accession number/rotavirus group/species origin/country of origin/G/P-types represent the reference sequences included in the tree. The viruses characterized in this study with their Bioproject accession number are shown in red text. The vertical lines represent genotypes.

**Figure 4 pathogens-12-01199-f004:**
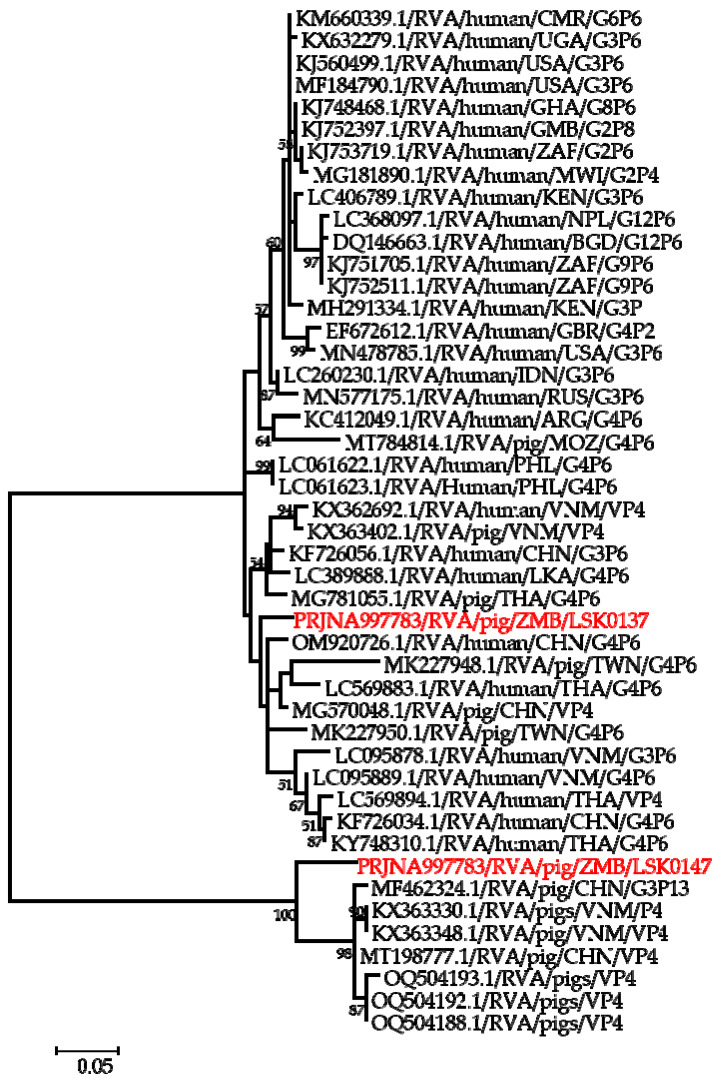
Phylogenetic tree of VP4 genes that belong to genotype P6. The tree was constructed using the general time reversible evolutionary model. The analysis was based on 2075 nucleotides. Bootstrap values on branch nodes ≥ 50% are shown. The GenBank accession number/rotavirus group/species origin/country of origin/G/P-types represent the reference sequences included in the tree. The virus characterized in this study with its Bioproject accession number is shown in red text.

**Figure 5 pathogens-12-01199-f005:**
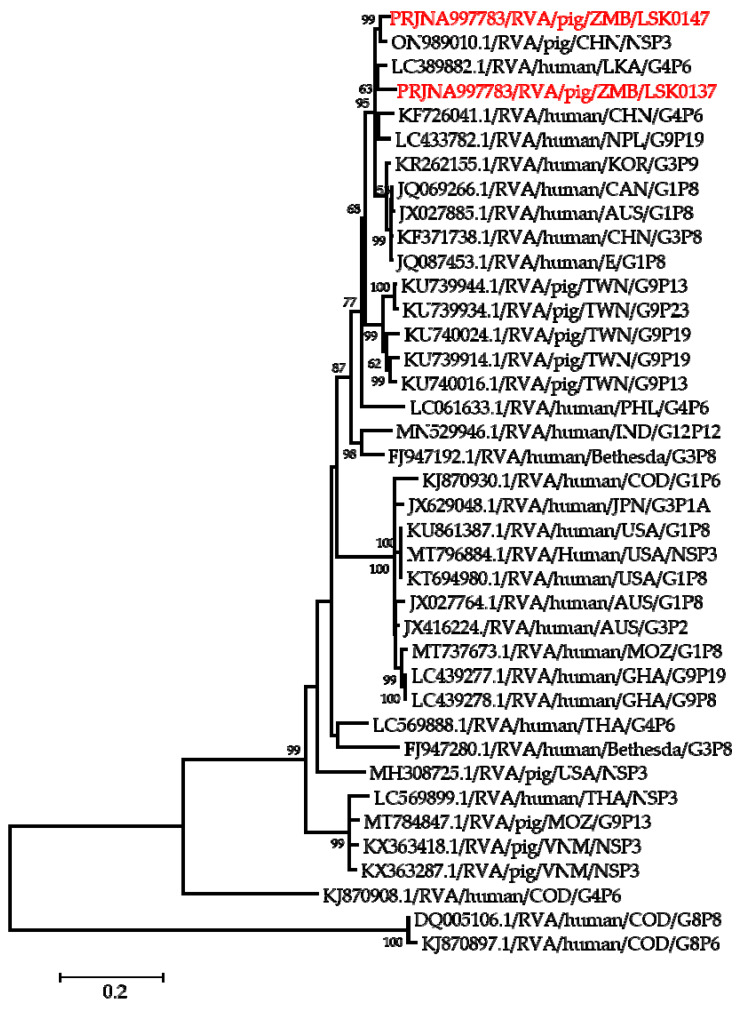
Phylogenetic tree of NSP3 genes that belong to genotype T1. The tree was constructed using the general time reversible evolutionary model. The analysis was based on 656 nucleotides. Bootstrap values on branch nodes ≥ 50% are shown. The GenBank accession number/rotavirus group/species origin/country of origin/G/P-types represent the reference sequences included in the tree. The viruses characterized in this study with their Bioproject accession number are shown in red text.

**Figure 6 pathogens-12-01199-f006:**
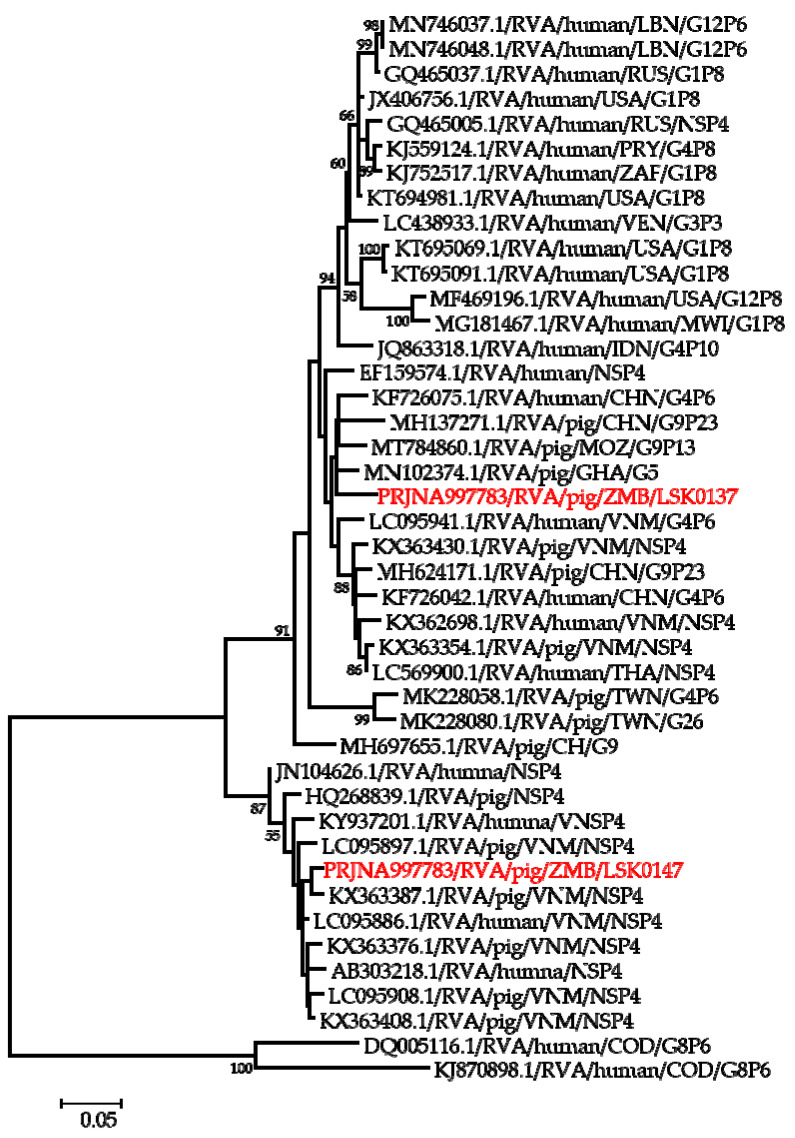
Phylogenetic tree of NSP4 genes that belong to genotype E1. The tree was constructed using the general time reversible evolutionary model. The analysis was based on 674 nucleotides. Bootstrap values on branch nodes ≥ 50% are shown. The GenBank accession number/rotavirus group/species origin/country of origin/G/P-types represent the reference sequences included in the tree. The virus characterized in this study with its Bioproject accession number is shown in red text.

**Table 1 pathogens-12-01199-t001:** Univariate logistic regression showing the selection of variables for multivariate regression.

Variable	Levels	PCR “+ve”	PCR “−ve”	Odds Ratio	95% CI	*p*-Value
Season	Dry	11	72	1		
	Rainy	23	42	3.58	1.62–8.34	0.00209
Diarrhea	No	6	41	1		
	Yes	28	73	2.62	1.06–7.47	0.0494
Age	Grower	1	9	1		
	Piglet	18	31	5.23	0.87–100.40	0.1310
	Weaner	15	74	1.82	0.31–34.91	0.5817

**Table 2 pathogens-12-01199-t002:** Multivariate logistic regression showing the dependence of rotavirus infection on various variables.

Variable	Levels	Odds Ratio	95% CI	*p*-Value
Season	Dry	1		
	Rainy	3.13	1.37–7.47	0.00781
Diarrhea	No	1		
	Yes	1.61	0.59–4.88	0.36649
Age	Grower	1		
	Piglet	4.18	0.62–83.71	0.20829
	Weaner	1.68	0.25–33.39	0.64534

**Table 3 pathogens-12-01199-t003:** Most identical genotypes of segment 9 of the VP7 genes of Zambian rotavirus detected in pigs and their percentage identities by BLASTn.

Sample ID	GenBank Accession No.	Stool		Closest Sequence		
			Country	Host	Genotype VP7 Gene	% Identity
14	OR294031	Diarrhea	China	Pig	G9	96.17
18	OR294032	Diarrhea	China	Human	G5	94.68
31	OR294033	Diarrhea	China	Pig	G9	96.38
32	OR294034	Diarrhea	China	Pig	G9	96.05
34	OR294035	Diarrhea	Ghana	Pig	G5	87.52
41	OR294036	Diarrhea	China	Pig	G9	96.42
42	OR294037	Diarrhea	China	Pig	G9	95.84
71	OR294038	Diarrhea	China	Human	G5	94.48
72	OR294039	Diarrhea	China	Human	G5	94.28
77	OR294040	Normal	China	Human	G5	94.01
137	OR294041	Diarrhea	China	Human	G4	97.1
147	OR294042	Diarrhea	Taiwan	Human	G9	92.2

**Table 4 pathogens-12-01199-t004:** Genotypes and similar strains from the Genbank of all the segments of samples LSK0137 and LSK0147 analyzed from Rotavirus A Genotyping tool Version 0.1 and BLASTn.

Sample ID	Bioproject Accession No.	Gene	Genotype	Cut off % Value	BLASTn % Identity	Similar Strain from GenBank
LSK0137	PRJNA997783	VP7	G4	80	97.24	RVA/Human-wt/CHN/2018/G4P[6]
LSK0137	PRJNA997783	VP4	P[6]	80	95.58	RVA/Human-wt/CHN/B24-R2/2019/P6 VP4
LSK0137	PRJNA997783	VP6	I5	85	96.02	RVA/Pig-wt/VNM/14150_53/VP6
LSK0137	PRJNA997783	VP1	R1	83	95.20	RVA/Human-tc/VNM/NT0042/2007/G4P[6]
LSK0137	PRJNA997783	VP2	C1	84	96.39	RVA/Human-wt/CHN/R1954/2013/G4P[6]
LSK0137	PRJNA997783	VP3	M1	81	95.74	RVA/Human-wt/RUS/2015 VP3
LSK0137	PRJNA997783	NSP1	A8	79	97.35	RVA/Human-tc/VNM/NT0042/2007/G4P[6]
LSK0137	PRJNA997783	NSP2	N1	85	96.76	RVA/Human-tc/VNM/NT0042/2007/G4P[6]
LSK0137	PRJNA997783	NSP3	T1	85	95.91	Human rotavirus A strain GX54
LSK0137	PRJNA997783	NSP4	E1	85	97.97	RVA/Pig/China/FJSH01/2021/G26P[23]
LSK0137	PRJNA997783	NSP5	H1	91	98.83	RVA/Human-wt/BRA/HST327/1999/G4P[6]
LSK0147	PRJNA997783	VP7	G9	80	94.22	RVA/Human-wt/TWN/G9P19
LSK0147	PRJNA997783	VP4	P[x]	80	93.15	RVA/Pig-wt/VNM/14150_54/VP4
LSK0147	PRJNA997783	VP6	I5	85	94.54	Porcine rotavirus strain JN-1 VP6
LSK0147	PRJNA997783	VP1	R1	83	96.97	RVA/Human-wt/THA/PK2015-1-0001 VP1
LSK0147	PRJNA997783	VP2	C1	84	92.25	RVA/Human-wt/LKA/R1207/2009/G4P[6]
LSK0147	PRJNA997783	VP3	M1	81	98.68	RVA/Pig-wt/VNM/14225_44/VP3
LSK0147	PRJNA997783	NSP1	Ax	79	96.31	RVA/Human-wt/CHN/E931/2008/G4P[6]
LSK0147	PRJNA997783	NSP2	N1	85	96.85	Porcine rotavirus A isolate GDJM1NSP2
LSK0147	PRJNA997783	NSP3	T1	85	97.60	Pig-wt/CHN/CN127/2021/G12P[7] NSP3
LSK0147	PRJNA997783	NSP4	E1	85	98.21	RVA/Human-tc/VNM/NT0001/2007/G3P[6]
LSK0147	PRJNA997783	NSP5	H1	91	99.58	RVA/Human-wt/LKA/R1207/2009/G4P[6]

RVA = rotavirus A; VP = viral proteins; NSP = nonstructural protein; X = not genotyped.

## Data Availability

The data presented in this study are available in this article and Appendix A. Sequence data generated in this study has been deposited in public databases and are freely available.

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
