# Peer review of "Prevalence and Genomic Characterization of Rotavirus A from Domestic Pigs in Zambia: Evidence for Possible Porcine–Human Interspecies Transmission"

_pathogens, 2023, doi:10.3390/pathogens12101199_

Round 1

Reviewer 1 Report

This study aimed to study the prevalence of RVA in fecal samples collected from pig farms in Zambia. The manuscript is well written, and the data is interesting.

1)      In this study a total of 148 samples collected from 6 farms were used. However, it remains unknown whether there was a difference in RVC prevalence among these farms (different location and commercial/smallholder).

2)      Similarly, it might be worth mentioning and discussing from what age group the positive samples were detected/characterized.

3)      A group of growers looks underrepresented in this study, which should be considered in the analysis.

4)      Please, specify what exact model was used for every phylogenetic tree.

 Line 293 – to RVA

Line 442 – porcine- pig

Reviewer 2 Report

The manuscript entitled “Prevalence and Genomic Characterization of Rotavirus A from Domestic Pigs in Zambia: Evidence for Porcine Human Interspecies Transmission” is well written, describing the genomic characterization of Rotavirus A strains from pigs in Zambia. In this year round cross-sectional study (during January 2018 – December 2018) different Rotavirus G- genotypes (VP7) have been identified with closest similarity with strains from both human and porcine origin, suggesting interspecies transmission. For the betterment of the manuscript some major and minor comments are as follows:

Major comments:

1. Line no 251 – 260, and table 3; Only 12 samples were VP7 positive (~ 35.3%) out of 34 VP6 positive samples, therefore 22 samples  (>64%) were VP7 negative. The authors should discuss here the possible reasons why these 22 VP6 positive samples failed to detect VP7 gene? Moreover, only VP7 gene were screened among the 34 VP6 positive samples, are the VP4 gene were screened among these 34 VP6 positive samples? As VP4 designates the P genotype of RV, which is also important to mention besides G (VP7) genotypes. The authors should discuss about the reasons not determining the P genotypes of RV strains in this study.   

2. Line no 327, in the section 3.4 “Phylogenic analysis of RVA Nonstructural Proteins Genes” the authors constructed phylogenetic trees for all NSPs (NSP1-NSP5) for LSK0137, but phylogenetic trees of only NSP2 and NSP3 were constructed for LSK0147. The authors should include the NSP4 and NSP5 genes of LSK0147 strain along with the phylogenetic analysis of NSP4 and NSP5 of LSK0137 stain.  

3. Line no 390-400, the genotypes of VP4 and NSP1 genotypes of LSK0147 strain were proposed on the basis of nucleotide similarity on sequence obtained from partial genomic segments. The author should mention here about nucleotide identity percentage (values in %) with those closest similar strains, eliminating the genotypes with lower nucleotide identity percentage. Construction of phylogenetic tree with partial sequences of VP4 and NSP1 is another possible way to support the proposal of probable genotypes of LSK0147 strain.

Minor comments:

1. Line no 104, “genotype 1, genotype 2 and genotype 3 referred to as Wa like, DS 1 like and AU 1 like respectively” the authors should change the designation from ‘genotype’ to ‘genogroup’ (like ‘genotype 1’ should be ‘genogroup 1’ and so forth) to avoid confusion between genotype constellation and genotype of individual gene segments. For further details please follow the following article.

[Matthijnssens J, Ciarlet M, Heiman E, Arijs I, Delbeke T, McDonald SM, Palombo EA, Iturriza-Gómara M, Maes P, Patton JT, Rahman M, Van Ranst M. Full genome-based classification of rotaviruses reveals a common origin between human Wa-Like and porcine rotavirus strains and human DS-1-like and bovine rotavirus strains. J Virol. 2008 Apr;82(7):3204-19. doi: 10.1128/JVI.02257-07. Epub 2008 Jan 23. PMID: 18216098; PMCID: PMC2268446.]

2. Line no 72 – 81, the authors should focus on the role of interspecies transmission of RV on the efficacy of RV vaccine in this section, as there are reports that interspecies transmission may have negative influence over RV vaccination.

3. Line no 396 – 398, authors please check the accession numbers of the two Zambia strains, as the accession numbers are same that is ‘LSK0137’.

4. Line no 383, authors please check the word ‘strainse’.

Minor editing required

Reviewer 3 Report

In the Manuscript ID: pathogens-2585448, entitled “Prevalence and Genomic Characterization of Rotavirus A from Domestic Pigs in Zambia: Evidence for Porcine-Human Interspecies Transmission” the authors describe the situation regarding RVA in pigs in a certain region of Zambia. Despite being a study performed entirely in pigs, the authors focused in the possibility of zoonosis, trying to discuss the impact in human population leaving in a second plane the impact in pig’s production/industry. I recommend that you delve a little deeper in this regard, in addition to the following comments:

The Introduction is clear and well written. I recommend interchanging the first and second paragraphs, because the manuscript is in pigs not in humans (despite the possible interspecies transmission).

Line 151: 4-8 weeks are the weaners

Section 2.2: The number of samples should be described here for each pig category.

Section 2.3: Accession numbers should be detailed here.

Section 2.4: Accession number or project number should be detailed here.

Section 3.1: First two sentences should be in Section 2.2.

Table 1 and table 2 results: It is not clear why performed two statistical analyses, why first univariate and then multivariate?

Line 253 and 254: This sentence is not clear, “and the Biotechnology Information (NCBI) database”?

Line 258: Did you use BLASTn or the Rotavirus Virus A Genotyping tool?

Figures 2, 3 and 4: Phylogenetic trees are shown very structured, like bootstrap consensus. I consider better to show the original tree with the genetic distances observed by branches length.

Section 3.4: Phylogenic should be Phylogenetic

Line 330: Vietnum should be Vietnam

Figure 5 and 6: Why the phylogenetic trees of some genes are main figures and some others are supplemental?

Lines 363-364: Why these prevalence trends suggest that the overall prevalence of porcine RVA infections in Africa might be considerably high?? Some of the prevalences showed are lower than the reported here

Line 383: strainse should be strains

Lines 397-398: The common names of the strains are missing in the complete names according to RCWG

Discussion: Despite the evidence of possible interspecies transmission between pigs and humans, the authors discussed a lot about the origin of the different strains, which in my opinion is not so evident from the analyses performed. I agree that the interspecies transmission could be possible, but as stated above, the origin of the strains is not so evident.

Round 2

Reviewer 2 Report

The authors have addressed all reviewer's comments . The introduction, methodology, results section and discussion has been improved considerably. The limitation of the study is also mentioned. 

Though the authors could not perform alternative tests to type the untypable samples , there are limited reports from Africam continent. Thus the manuscript may be accepted

Author Response

Thanks you so much, for your response